# Meta-Learning for Graphs with Heterogeneous Node Attribute Spaces for Few-Shot Edge Predictions

**Zhong Chuang**                                                    *zhong.chuang.yw3@is.naist.jp*
*Division of Information Science*
*Nara Institute of Science and Technology*

**Yusuke Tanaka**                                                    *ysk.tanaka@ntt.com*
*NTT Communication Science Laboratories, Kyoto, Japan*

**Tomoharu Iwata**                                          *tomoharu.iwata.gy@hco.ntt.co.jp*
*NTT Communication Science Laboratories, Kyoto, Japan*

**Reviewed on OpenReview:** *https://openreview.net/forum?id=XXXX*

## Abstract

Prediction of edges between nodes in graph data is useful for many applications, such as social network analysis and knowledge graph completion. Existing graph neural network-based approaches have achieved notable advancements, but encounter significant difficulty in building an effective model when there is an insufficient number of known edges in graphs. Although some meta-learning approaches were introduced to solve this problem, having an assumption that the nodes of training graphs and test graphs are in homogeneous attribute spaces, which limits the flexibility of applications. In this paper, we proposed a meta-learning method for edge prediction that can learn from graphs with nodes in heterogeneous attribute spaces. The proposed model consists of attribute-wise message-passing networks that transform information between connected nodes for each attribute, resulting in attribute-specific node embeddings. The node embeddings are obtained by calculating the mean of the attribute-specific node embeddings. The encoding operation can be repeated multiple times to capture complex patterns. The attribute-wise message-passing networks are shared across all graphs, allowing knowledge transfer between different graphs. The probabilities of edges are estimated by the Euclidian distance between node embeddings. Experimental results on 14 real-world data sets demonstrate that the proposed method outperforms existing methods in edge prediction problems with sparse edge information.

## 1 Introduction

Many real-world data can be described by graphs, such as social networks, knowledge networks, and protein-protein interactions. The nodes of graphs represent individuals, and edges denote the relations or interactions between nodes. Among many graph-related research problems, edge prediction is an essential problem that attempts to estimate the probability of the edges between two nodes. For example, edge prediction has been proven useful in discovering new protein-protein interactions, which are costly for blindly checking all possible interactions. In social networking services, edge predictions are used to search for users' potential friends and provide recommendations.

Despite the early similarity-based and likelihood-based approaches, graph neural network-based approaches have received significant attention for their high performance on edge prediction tasks. The recent examples of Graph Convolutional Networks (GCN) applications include social networks (Rozemberczki & Sarkar, 2021) and knowledge graphs (Liu et al., 2021). GCN obtains node embeddings containing the graph's topology and node attribute information. It transforms node attributes—such as usage statistics of the

social network members, the belonging species of proteins, and bag-of-words of the articles—into a latent space and aggregates messages from neighboring nodes. The node embeddings are then used to predict edges based on similarities. Nonetheless, GCN requires a tremendous number of observed edges for training (Ding et al., 2022; Guo et al., 2021), making them inapplicable for many real-world applications, such as newly released services or when data collection is costly.

Some meta-learning approaches for graph convolutional networks have been used in previous works (Mandal et al., 2021; Huang & Zitnik, 2020) to solve the insufficient data issue. With the other accessible training graphs, these meta-learning approaches try to improve edge prediction performance even when there is sparse edge information from test graphs. However, these meta-learning approaches require all training and test graphs to share a common attribute space. This assumption limits the models' ability to learn from diverse graphs, potentially missing valuable insights from graphs with nodes in heterogeneous spaces. Additionally, if the node attributes of the test graphs are unknown, these methods become inapplicable as it is impossible to collect training data.

This research aims to propose a meta-learning model that can eliminate the limitation of node attribute spaces and achieve better performance in few-shot graph edge prediction problems. To fulfill this objective, we propose Heterogeneous Graph Meta-Learning (HGML), which can learn from graphs with nodes in heterogeneous attribute spaces and improve edge prediction performance on unseen graphs with sparse edge information.

HGML comprises multiple layers of attribute-wise message-passing networks with average pooling. Attribute-wise message-passing networks transform information from connecting nodes for each attribute to obtain attribute-specific node embeddings. The attribute-wise message-passing operation captures information from neighboring nodes and transforms graphs into a common latent space, accommodating graphs with heterogeneous node attributes. Since attribute-wise message-passing does not account for relationships between node attributes, we derive the node embeddings from the attribute-specific node embeddings using average pooling. The average pooling allows the inclusion of all attribute-wise information of nodes regardless of the number of attributes. By performing these two steps alternately, the node embeddings can include information from further nodes. The attribute-wise message-passing networks are shared for all graphs so the model can learn the common knowledge across different graphs. Unlike iterative updating procedures required in gradient-based meta-learning methods, such as model-agnostic meta-learning (Finn et al., 2017), or other encoder-decoder style meta-learning methods, such as neural processes (Garnelo et al., 2018b), which adapt to various tasks by only optimizing the parameters of neural networks using large training datasets, HGML can directly predict on new input graphs without local adaptations. We predict edge probabilities by Euclidean distances between node embeddings. The model parameters are trained by minimizing the expected test edge prediction loss across graphs with sparse edge information. The contributions of this research are summarized as follows:

- We propose a graph neural network-based model where the parameters can be shared among graphs with nodes in heterogeneous attribute spaces.

- We propose a meta-learning framework using our model to improve edge prediction performance with sparse edge information by learning from various graphs without bilevel gradient-based optimization.

- We experimentally demonstrate that HGML outperforms existing meta-learning methods for edge prediction using 14 real-world graphs with nodes in heterogeneous spaces.

This paper is organized as follows: Chapter 2 reviews previous research on the edge prediction problem of graphs and existing approaches, including graph convolutional networks, and meta-learning on graph convolutional networks. Chapter 3 introduces the proposed method, including the problem setting, the graph convolutional network-based model, and the meta-learning algorithm. Chapter 4 explains the details of the experimental setup, such as the dataset, comparison benchmarks, and the setup of training parameters. Chapter 5 presents the experimental results and discusses the effectiveness of the proposed method. Chapter 6 concludes with the contributions and limitations of this research and discusses potential model extensions for future works.

## 2 Related work

For edge prediction, many machine learning-based methods have been proposed. Started from graph convolutional networks (Kipf & Welling, 2017), the message passing-base methods have received significant results (Hamilton et al., 2018). These methods are used in various applications, which include the knowledge graph completion (Ye et al., 2022) and the discovery of protein-protein interactions (Zhou et al., 2022). However, these message-passing-based approaches require a lot of edge information for training.

Some meta-learning approaches have been proposed to solve the issue of lacking training data. For instance, by splitting the training graphs into many subgraphs, the model can learn to predict new graphs with a few edges by subgraph sampled from the original graph (Huang & Zitnik, 2020). Another approach is to perform model-agnostic meta-learning (Finn et al., 2017) on graph variational networks with an additional graph signature component for faster adaptation on graphs with a small number of edges (Bose et al., 2020). However, these approaches can not handle graphs in heterogeneous attribute spaces as the inputs.

A method for learning graphs in heterogeneous attribute spaces has been proposed (Hassani, 2022), where attributes in different graphs are padded to a fixed dimension. However, this method is only applicable when attributes of all training and test graphs are known at the meta-training phase. On the other hand, the proposed method in this paper can predict the edges of unseen graphs that are not provided at the meta-training phase.

To extend the flexibility of GNNs, the multi-domain generalized graph meta-learning (Lin et al., 2023) can be trained with graphs in heterogeneous attribute spaces and make predictions on graphs in unknown attribute spaces. However, the bilevel optimization for the task-wise neural networks and prediction model makes it computationally expensive and hard to train. In contrast to MAML-like meta-learning approaches, which learn initial meta-parameters for local adaptation, another meta-learning approach—based on encoder-decoder architectures (Garnelo et al., 2018a) uses neural networks to encode data, enabling the prediction of new test tasks by meta-learning a single set of parameters across diverse input graphs without requiring bilevel optimization. This kind of meta-learning approach optimizes the neural network to approximate the effect of fine-tuning. Although encoder-decoder meta-learning methods for tabular data have been proposed (Iwata & Kumagai, 2020), they are not applicable for edge prediction in graph data.

Another solution is transfer learning(or domain adaptation), by learning the relationship between the source and target domain, the model can learn to predict on test graphs even when few edges are available (Zheng et al., 2023; Mallick et al., 2020). However, these approaches can only be applied to a pair of training and test graphs, and both the training and test graphs have to be available during the training phase. Different from the transfer learning approaches, we propose a model that can learn from multiple training graphs and make predictions on multiple test graphs even when they are not available for training.

## 3 Method

### 3.1 Problem formulation

We illustrate the meta-learning process from graphs in heterogeneous attribute spaces in Figure 1. In the training phase, given multiple original graphs with nodes in heterogeneous attribute spaces $\mathbf{G} = \{\bar{\mathcal{G}}_d\}_{d=1}^D$, where $\bar{\mathcal{G}}_d = (\bar{\mathbf{X}}_d, \bar{\mathbf{A}}_d)$ is the $d$th graph with $\bar{N}_d$ nodes and $\bar{I}_d$ attributes. For any two original graphs $\bar{\mathcal{G}}_d$ and $\bar{\mathcal{G}}_{d'}$, their attribute spaces and sizes may be different, $\bar{I}_d \neq \bar{I}_{d'}$, and their node sizes may be different $\bar{N}_d \neq \bar{N}_{d'}$. Although we assume undirected graphs for simplicity, the proposed method is straightforwardly applicable to directed graphs.

For each training step, the model is trained by the training graphs $\{\mathcal{G}_t\}_{t=1}^{\mathcal{T}}$, which are formed by sampling nodes and the corresponding edges from $\mathbf{G}$. $\mathcal{G}_t = (\mathbf{X}_t, \mathbf{A}_t)$ have $N_t$ nodes and $I_t$ attributes, where $\mathbf{X}_t \in \mathbb{R}^{N_t \times I_t}$ is the attribute matrix, in which the rows represent the nodes, and the columns indicate the node attributes. $\mathbf{A}_t \in \{0,1\}^{N_t \times N_t}$ is the adjacency matrix where the rows and columns represent the nodes. A value of one indicates the presence of an edge between two nodes, while zero indicates the absence of an edge. For simplicity, we will ignore the subscript $t$ in the following. In the test phase, a test graph $\mathcal{G}^* = (\mathbf{X}^*, \mathbf{A}^*)$

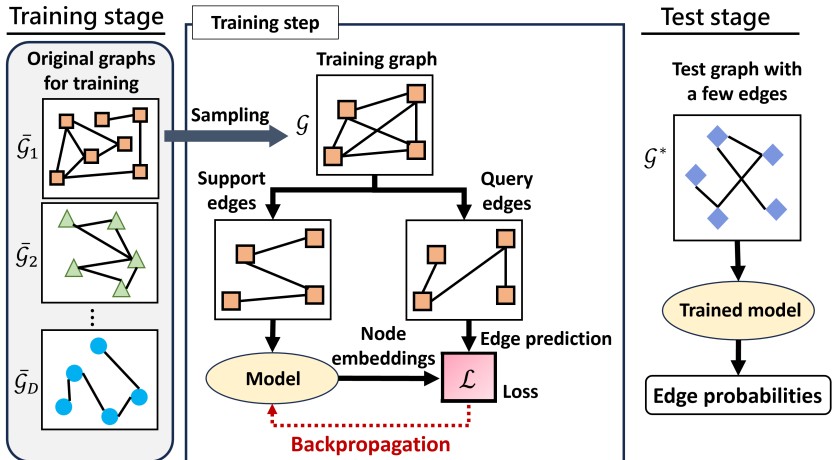

Figure 1: The meta-learning process of HGML.

with sparse edge information is given, where $\mathcal{G}^*$ is different from any $\bar{\mathcal{G}}_d$ and its node attributes differ from those in all training graphs. This research aims to predict the probability of edges in the test graph.

## 3.2 Model

HGML predicts the probability of edges between nodes from attribute matrix $\mathbf{X}$ and adjacency matrix $\mathbf{A}$ with $N$ nodes and $I$ attributes. In the training phase, $\mathbf{X}$ and $\mathbf{A}$ are generated by sampling a subgraph from training graphs. In the test phase, they correspond to $\mathbf{X}^*$ and $\mathbf{A}^*$ of a test graph.

We demonstrate HGML in Figure 2. HGML is implemented by repeatedly passing graphs into attribute-wise message-passing networks along with average pooling. With the attribute-wise message-passing networks and average pooling, we can obtain node embeddings containing node attributes and graph structure information, enabling HGML to learn graphs' common knowledge even when attribute spaces differ. The attribute-wise message-passing networks can be performed multiple times to learn from further neighbors. After transforming the node embeddings into a common space, we calculate the probability of edges using the final node embeddings.

### 3.2.1 Attribute-wise message-passing networks

The attribute-specific node embedding of the $n$th node's $i$th attribute at the $\ell$th encoding layer $\mathbf{v}_{ni}^\ell$ is calculated using the $i$th attribute values of the $n$th node and its neighboring nodes. Starting from the second layer, instead of the attribute values, we input the concatenation vector of attribute-specific node embeddings and node embeddings from the previous layer to the attribute-wise message-passing networks.

$$\mathbf{v}_{ni}^\ell = \begin{cases} \frac{1}{|\mathcal{N}_n|}\left(\sum_{m\in\mathcal{N}_n} f_{\mathrm{v}}^\ell(x_{mi})\right) & \text{for } \ell = 1 \\ \frac{1}{|\mathcal{N}_n|}\left(\sum_{m\in\mathcal{N}_n} f_{\mathrm{v}}^\ell([\mathbf{v}_{mi}^{\ell-1}, \mathbf{z}_m^{\ell-1}])\right) & \text{for } \ell = 2,\dots,L, \end{cases} \tag{1}$$

where $x_{mi} \in \mathbb{R}$ is the $i$th attribute of the $n$th node, $f_{\mathrm{v}}^\ell : \mathbb{R} \to \mathbb{R}^{K_{\mathrm{v}}^\ell}(\text{for } \ell = 1), \mathbb{R}^{K_{\mathrm{v}}^\ell} \to \mathbb{R}^{K_{\mathrm{v}}^\ell}(\text{for } \ell \geq 2)$ are Feed-forward Neural Networks (FNNs), $K_{\mathrm{v}}^\ell$ is the output channel of FNNs at $\ell$th layer, $\mathcal{N}_n$ is the index set of the neighboring nodes and $n$th node itself, $|\cdot|$ indicates the element count in the set, $[\cdot,\cdot]$ is the concatenate operation, $L$ is the numbers of encoding layers, and $\mathbf{z}_n^{\ell-1}$ is the node embedding of the $n$th node in the previous layer, which is calculated by taking the mean across attributes:

$$\mathbf{z}_n^\ell = \frac{1}{I}\sum_{i=1}^{I}\mathbf{v}_{ni}^\ell. \tag{2}$$

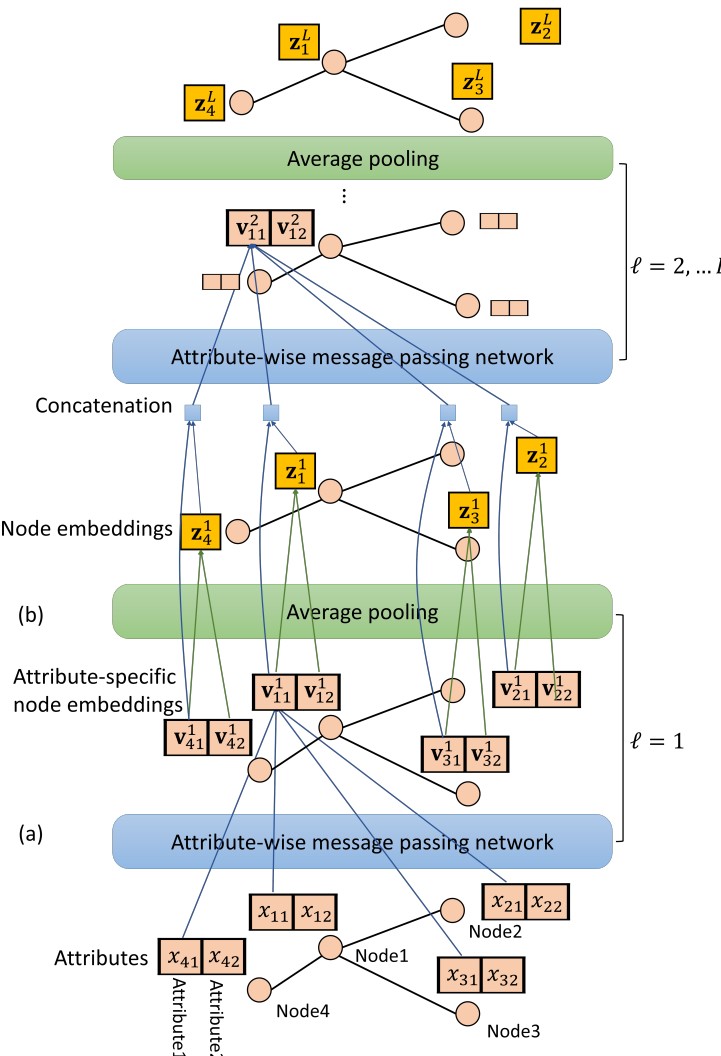

Figure 2: attribute-wise message-passing networks' forward passing in HGML. The attribute-wise message-passing networks can perform message-passing regardless of the number of attributes, and the average pooling operation aggregates the relationships between these attributes.

For $\ell = 1$, Eq. (1) represents the sum of transformed messages from the neighbors and the second term accounts for the node's self-message. Eq. (1) is based on message-passing networks (Kipf & Welling, 2017). However, unlike the existing networks, the attribute-wise message-passing networks can learn from graphs with various numbers of attributes with a shared neural network due to the attribute-wised operation. The average operation across attribute-specific node embeddings as Eq. (2) not only allows the model to learn the relationship between attributes regardless of the number of attributes but makes HGML a permutation invariant network (Zaheer et al., 2017), which ensures that the model performs invariant even when node attributes are permuted.

### 3.2.2 Edge prediction

The probability of edge existence between the $n$th and the $m$th nodes is represented as $\hat{a}_{nm}$, which can be estimated by the Euclidian distance between node embeddings:

$$\hat{a}_{nm}(\mathbf{X}, \mathbf{A}, \mathbf{\Phi}) = \exp(-||\mathbf{z}_n - \mathbf{z}_m||^2), \tag{3}$$

where $\boldsymbol{\Phi}$ represents the parameter set of the neural networks: $f_v^\ell$ for $\ell = 1, 2, \ldots, L$. $\mathbf{z}_n$ and $\mathbf{z}_m$ are node embeddings, and $||\cdot||$ indicates the L2 norm. We use the norm operation to evaluate the Euclidean distance between two node embeddings. The closer the node embeddings are, the higher the probability that an edge exists between them. The edge probabilities are constrained to be between zero and one due to the negative exponent.

### 3.3 Meta-learning

We train the model by maximizing the following smoothed area under the ROC curve (AUC) (Hanley & McNeil, 1982):

$$J(\mathbf{A}^q|\mathbf{X}, \mathbf{A}^s; \Phi) = \frac{1}{|\mathbf{A}^{q^+}|} \frac{1}{|\mathbf{A}^{q^-}|} \sum_{n,m \in \mathbf{A}^{q^+}} \sum_{n',m' \in \mathbf{A}^{q^-}} \sigma(-||\mathbf{z}_n - \mathbf{z}_m||^2 + ||\mathbf{z}_{n'} - \mathbf{z}_{m'}||^2), \tag{4}$$

where $\mathbf{A}^s$ is an adjacency matrix with support edges that are used to obtain node embeddings, $\mathbf{A}^q$ is an adjacency matrix with query edges that are different from support edges and used to calculate the model's evaluation loss, $\mathbf{A}^{q^+}$ indicate the node pairs with edges linked in $\mathbf{A}^q$, $\mathbf{A}^{q^-}$ are the negative samples, representing node pairs without edges linked in $\mathbf{A}^q$. By maximizing Eq. 4, we make the distances between nodes with linked edges close to zero, when pulling the unlinked nodes as far as possible.

Numerous objective functions, such as binary cross-entropy or other pair-wised loss functions (Zhu et al., 2022), are commonly used to optimize classification models. However, it has been shown that objective functions designed to minimize misclassification error may not necessarily maximize the AUC score (Cortes & Mohri, 2003). Similar to the previous work (Iwata & Yamanaka, 2019), which applied the smoothed AUC ROC score as the objective function for an imbalanced classification problem. Consequently, we estimate the model parameters $\boldsymbol{\Phi}$ with original graphs $\bar{\mathcal{G}}_d$ by following expected test edge prediction smoothed AUC:

$$\hat{\boldsymbol{\Phi}} = \underset{\boldsymbol{\Phi}}{\arg\max}\ \mathbb{E}_{d \sim D}[\mathbb{E}_{\mathcal{G} \sim \bar{\mathcal{G}}_d}[\mathbb{E}_{\mathbf{A}^q \sim \mathcal{G}}[J(\mathbf{A}^q|\mathbf{X}, \mathbf{A}^s; \boldsymbol{\Phi})]]], \tag{5}$$

where $\mathbb{E}_{d \sim D}$ is the expectation over all original graphs, $\mathbb{E}_{\mathcal{G} \sim \bar{\mathcal{G}}_d}$ is the expectation over all sampled training graphs and $\mathbb{E}_{\mathbf{A}^q \sim \mathcal{G}}$ is the expectation for edges in $\mathbf{A}^q$. The expectation is calculated by Monte Carlo sampling. We illustrate the proposed meta-learning model as Algorithm 1. In line 2, we reset the loss and its gradient as the beginning of an epoch. In lines 4-5, we randomly pick one of the original graphs $\bar{\mathcal{G}}_d$, uniformly random sample $N$ nodes, and $I$ attributes for a training graph $\mathcal{G} = (\mathbf{X}, \mathbf{A})$. We sample the subgraphs with replacement and the order of the sampled attributes is randomly shuffled. The isolated nodes of the subgraphs are removed. In line 6, we randomly split the edges in $\mathcal{G}$ with support edge ratio $r$ to generate support adjacency matrix $\mathbf{A}^s$ and query adjacency matrix $\mathbf{A}^q$. In line 7, we input $(\mathbf{X}, \mathbf{A})$ into HGML's forward pass as described in Figuire 2 for estimating edge probabilities $\{\hat{a}_{nm}\}_{n,m=1}^N$. In line 8, we calculate the loss with Eq. (4) and add it to loss $\mathcal{L}$. We repeat lines 3-9 for accumulating the loss and its gradient to $\mathcal{L}$ as a batch. In line 10 we update the model parameters with learning rate $\gamma$.

Although we did not impose specific conditions on the input graphs, we assumed an underlying common feature distribution across them. The proposed meta-learning algorithm in HGML enables the model to meta-learn these common patterns from the input graphs.

## 4 Experimental Evaluation

### 4.1 Datasets

We evaluated our model with 14 real-world original graphs from 12 datasets: Amazon, Blogcatalog, Cora, CiteSeer, Coauthor, DeezerEurope, DBLP, Facebook, PubMed, Reddit, Twitch, and WikiCS.

Amazon (Shchur et al., 2018) includes two segments of the Amazon co-purchase graphs, where the nodes represent goods, the edges indicate that two goods are frequently bought together, and the node attributes are bag-of-words encoded product reviews. This dataset has two independent graphs representing two categories of products: computers and photos.

---

**Algorithm 1** Training procedure of proposed model: $Subgraph(\mathcal{G}, N, I)$ randomly sample subgraph of $N$ nodes and $I$ attributes. $RandomSplit(\mathbf{A}, r)$ generate support and query adjacency matrix $\mathbf{A}^{\mathrm{s}}$, $\mathbf{A}^{\mathrm{q}}$ by randomly splitting edges from $\mathbf{A}$ with support edge ratio $r$.

---

**Input**: original graphs $\{\bar{\mathcal{G}}_d\}_{d=1}^{D}$, with batch size $B$, sample size $N$, number of sample attributes $I$, and sampling rate of support adjacency matrix $r$, learning rate $\gamma$
**Output**: Trained model parameters $\mathbf{\Phi}$

1: **while** not done **do**
2:     Initialize loss, $\mathcal{L} \leftarrow 0$.
3:     **for** $1, 2, ..., B$ **do**
4:         Select graph $d$ from $\{1, 2, ..., D\}$.
5:         Sample training graph $\mathcal{G} \leftarrow Subgraph(\bar{\mathcal{G}}_d, N, I)$.
6:         Sample support adjacency matrix $\mathbf{A}^{\mathrm{s}}, \mathbf{A}^{\mathrm{q}} \leftarrow RandomSplit(\mathbf{A}, r)$.
7:         Predict the edge probabilities $\{\hat{a}_{nm}\}_{n,m=1}^{N}$ by input $\mathbf{X}, \mathbf{A}^{\mathrm{s}}$ as described in Section 3.2.
8:         Calculate loss by Eq. (4), $\mathcal{L} \leftarrow \mathcal{L} + J(\mathbf{A}^{\mathrm{q}}|\mathbf{X}, \mathbf{A}^{\mathrm{s}}; \mathbf{\Phi})$.
9:     **end for**
10:     Update model parameters $\mathbf{\Phi}$ with loss $\mathcal{L}$ and its gradient with learning rate $\gamma$.
11: **end while**

---

BlogCatalog (Yang et al., 2020) is a blogger-blogger interaction social network, where the nodes represent users and the edges represent friendship relationships. The node attributes are bag-of-words vectors containing the keywords of user profiles.

Cora (McCallum et al., 2000) is a citation network for scientific publications. The nodes are publications, and the edges represent their reference relationship. Each node is attributed by a 0/1-valued word vector indicating the absence/presence of the corresponding word from the dictionary. The dictionary consists of 1,433 unique words.

CiteSeer (Giles et al., 1998) is a citation network for scientific publications. The nodes represent the scientific publications, and the edges indicate the citation relationships between these publications. The node attributes are 0/1-valued word vectors, indicating the absence or presence of the corresponding word from the dictionary, which consists of 3,703 unique words.

Coauthor (Shchur et al., 2018) are two co-author networks based on the Microsoft Academic Graph. Nodes are authors, edges represent the co-authorship of papers, and node attributes are 0/1-valued word vectors representing the keywords for each paper. It includes two graphs from different research fields: Physics and Computer Sciences (CS).

DeezerEurope (Rozemberczki & Sarkar, 2020) is a social network source from the music streaming service: Deezer. The nodes represent users, and the edges are the friendships between users. The node attributes are the features associated with users, which can be derived from their activities and preferences.

DBLP (Pan et al., 2016) is a citation network of papers in the computer sciences. The nodes represent the research papers, and the edges indicate the citation relationships between these papers. The node attributes are bag-of-words representations of the papers' contents.

Facebook (Rozemberczki et al., 2021) is a network of Facebook pages where nodes correspond to pages and edges correspond to mutual likes between these pages. The 128 numerical node attributes include a 100-dimensional word embedding summarizing the page's textual content, activity-related information, engagement metrics, and metadata.

Reddit (Hamilton et al., 2017) is a post-to-post graph dataset from Reddit posts made in September 2014. The nodes stand for posts and the edges exist if a user comments on both posts. For each post, two 300-dimensional GloVe CommonCrawl word vectors (Pennington et al., 2014) are used to represent the average word embeddings of the post title and the post's comments. Adding on the post's score and the number of comments made, there are 602 attributes in total.

Table 1: Statistics of the graph datasets.

| Graph | Network type | Attribute type | #Nodes | #Edges | #Attr. |
|-------|-------------|----------------|--------|--------|--------|
| AmazonComputers | Co-purchase | Bag-of-words | 13,471 | 491,722 | 500 |
| Blogcatalog | Social network | Bag-of-words | 5196 | 343,486 | 500 |
| AmazonPhoto | Co-purchase | Bag-of-words | 7,535 | 238,162 | 500 |
| DBLP | Paper reference | Bag-of-words | 17,716 | 105,734 | 500 |
| Coauthor CS | Co-authorship | Bag-of-words | 18,333 | 163,788 | 500 |
| Coauthor Physics | Co-authorship | Bag-of-words | 19,661 | 170,862 | 500 |
| DeezerEurope | Social network | Numerical | 28,281 | 185,504 | 128 |
| Facebook | Pages-likes-pages | Numerical | 22,470 | 341,646 | 128 |
| CiteSeer | Paper reference | Bag-of-words | 3,279 | 9,104 | 500 |
| Cora | Paper reference | Bag-of-words | 2,708 | 10,556 | 500 |
| PubMed | Paper reference | Bag-of-words | 19,717 | 88,648 | 500 |
| Reddit | Post-to-post | Word embeddings | 26,936 | 1,655,412 | 500 |
| Twitch | Social network | Numerical | 7,126 | 70,648 | 128 |
| WikiCS | Page reference | Word embeddings | 11,364 | 431,206 | 300 |

Twitch (Rozemberczki & Sarkar, 2021) is a user-user network connected by mutual friendships, and attributes of games like, location, and streaming habits are included.

WikiCS (Mernyei & Cangea, 2020) is an articles network derived from Wikipedia, where the nodes represent Wikipedia articles, and the edges represent hyperlinks between articles. The node attributes are multi-dimensional feature vectors that encapsulate the textual content of Wikipedia articles.

We uniformly sampled 30,000 nodes randomly for both the Reddit and CoauthorPhysics datasets. To avoid sparse attributes, we only used the top 500 frequency words for bag-of-words for AmazonComputers, AmazonPhoto, Blogcatalog, DBLP, CoauthorCS, CoauthorPhysics, CiteSeer, Cora, and Reddit. We normalized all attributes to the range from zero to one. All graphs' edges were considered to be undirected. The statistics of these original graphs are listed in Table 1.

## 4.2 Task

We randomly selected four original graphs, which were not used for training or validation, and sampled 100 subgraphs as test graphs. Each test graph was created by uniformly sampling 500 nodes and 70 attributes from the original graphs. For each sampling iteration, we replaced the sampled nodes and attributes, allowing the nodes and attributes to repeat across test graphs. We randomly shuffled the order of attributes in the sampled test graphs to eliminate any ordering information. The statistics of the sampled subgraphs are listed in Table 2.

A portion of the graph edges is designated as known support edges, while the remaining edges are treated as query edges. The objective of the experiment is to predict the query edges in the test graphs by applying meta-learning on the remaining original graphs. We validated HGML and the baseline methods using three different support edge ratios $r = \{0.3, 0.5, 0.7\}$ for various edge density situations.

## 4.3 Experimental Setup

We sampled subgraphs as training graphs to increase graph diversity, including variations in topological properties and combinations (or orders) of node attributes. The model can also be trained without sampling, but in that case, the training graphs would need to inherently exhibit diverse topological properties and node attributes. We generated training graphs with the number of nodes $N = 500$ and attributes $I = 70$ from eight original graphs, which are not used for creating test graphs. The generation of training graphs is the same as that of the test graphs, the statistics of the sampled subgraph can refer to Table 2, as well.

Table 2: Statistics of the sampled subgraphs. The reported values represent the mean and standard deviation derived from 100 sampling iterations.

| Graph | #Nodes | #Edges | #Attr |
|---|---|---|---|
| AmazonComputers | 281.40±17.46 | 349.90±60.93 | 70 |
| AmazonPhoto | 345.50±16.40 | 521.53±72.17 | 70 |
| Blogcatalog | 484.37 ± 3.67 | 1,573.70 ± 97.85 | 70 |
| DBLP | 69.27±9.32 | 42.77±7.69 | 70 |
| CoauthorCS | 98.13±12.77 | 60.37±10.09 | 70 |
| CoauthorPhysics | 86.60±11.91 | 52.33±7.84 | 70 |
| DeezerEurope | 54.43±11.14 | 30.87±7.05 | 70 |
| Facebook | 113.53±13.49 | 86.13±11.91 | 70 |
| CiteSeer | 164.00±13.77 | 108.03±13.21 | 70 |
| Cora | 234.93±14.70 | 179.27±19.59 | 70 |
| PubMed | 50.43±10.39 | 29.67±6.88 | 70 |
| Reddit | 230.00±15.25 | 282.67±52.33 | 70 |
| Twitch | 182.13±26.76 | 177.33±42.50 | 70 |
| WikiCS | 256.70±20.54 | 423.37±63.36 | 70 |

Three-layered feed-forward neural networks are used to $f_v^\ell$. The model encoder layer $L$ is set to three, and the number of hidden units and output channels $K_v^\ell$ are set to 64 for all $\ell$. Rectified linear units, $\text{ReLU}(x) = \max(0, x)$, are applied as activation. Optimization is performed using Adam (Kingma & Ba, 2015), with learning rate $\gamma = 10^{-3}$ and dropout rate $10^{-1}$. We train the model for 30,000 epochs. The batch size $B$ is set to 50.

The validation graphs were used for early stopping. We sampled 100 validation graphs from two original graphs with the same method as creating the test graphs. The original graphs used to create the validation graphs are not used for generating testing or training graphs. We evaluated the model performance using the area under the ROC curve (AUC) (Hanley & McNeil, 1982) by calculating the mean score across all test graphs. The selection of original graphs for generating training, validation, and test graphs was redone each time. We ran 10 experiments for each setting and took the mean results. The experiments were implemented with Pytorch (Paszke et al., 2017) in Python 3.11.3 and conducted on a computer with Xeon Platinum 8180 2.5GHz CPU, Nvidia Volta100, and 384GB memory, running the 64-bit Ubuntu 22.04.1 LTS.

## 4.4 Comparative methods

We compared HGML with five baseline methods. For the meta-learning method, We combined DeepSets (Zaheer et al., 2017) and GCN (DSGCN), and MAMLGCN. MAMLGCN is a single method that applies model-agnostic meta-learning (MAML) to GCN. We used a standard GCN as the prediction model and optimized the meta-parameters through a few task-specific local adaptations, following the original MAML approach. Because MAMLGCN is a meta-learning model that can not work on graphs with nodes in heterogeneous attribute spaces, We trained MAMLGCN without using node attributes. For non-meta-learning comparative methods, Neural Networks (NN), GCN, and Graph ATtention networks (GAT) (Veličković et al., 2018) were trained for each test graph only with the support edges. We did not implement the fine-tuned version of the comparative methods because the previous work (Iwata & Kumagai, 2020) similarly encodes attribute-specific information into a latent space and decodes it into prediction values. Their results demonstrated that this encoder-decoder meta-learning approach can outperform existing approaches that rely on fine-tuning.

For all neural network-based benchmark approaches, we set the number of hidden units and output channels to 64, the dropout rate to $10^{-1}$, and used rectified linear units, $\text{ReLU}(x) = \max(0, x)$, as activation function. The edge probabilities between any node pair $\{n, m\}$ for the comparative methods are estimated using the inner product of the node embeddings. DSGCN contains two neural network components: the feature extractor and GCN. First, we input the node attribute vectors into the feature extractor to convert the

Table 3: Performance of HGML and the comparable methods under different support edge rates $r$. Bold text indicates the highest mean AUC scores. The asterisk (*) denotes results significantly higher than all other comparative methods, as determined by a paired t-test with a p-value of less than 0.05.

| Model | $r = 0.3$ | $r = 0.5$ | $r = 0.7$ |
|---|---|---|---|
| **HGML** | **0.633±0.031** | **0.687±0.025*** | **0.719±0.050*** |
| DSGCN | 0.519±0.019 | 0.526±0.058 | 0.584±0.035 |
| MAMLGCN | 0.492±0.044 | 0.538±0.036 | 0.620±0.015 |
| NN | 0.590±0.013 | 0.593±0.010 | 0.605±0.016 |
| GCN | 0.604±0.017 | 0.603±0.023 | 0.637±0.027 |
| GAT | 0.617±0.022 | 0.618±0.028 | 0.652±0.027 |

attributes to a latent space $\mathbf{x}'_n = g\left(\frac{1}{I}\sum_{i=1}^{I} f(x_{ni})\right)$. $x_{ni}$ is the $i$th attribute of the $n$th node, and $g, f$ are FNN with three hidden layers. Then, we input the converted attributes and the adjacency matrix into the GCN to obtain the node embeddings. We trained DSGCN for 30,000 epochs with the learning rate $10^{-3}$, and the parameters were updated every batch for 50 training graphs. The training graphs are resampled every epoch. MAMLGCN is comprised of two graph convolutional layers and three linear output layers. We trained MAMLGCN without using node attributes because MAML can not deal with attributes in heterogeneous spaces. Unlike HGML and DSGCN, we trained the MAMLGCN with 50 training graphs without resampling every epoch. We set the local adaptation steps to 10, the learning rate $10^{-3}$ for the local adaptation, and the learning rate $10^{-3}$ for meta-parameters optimization. We trained MAMLGCN for 500 epochs. NN contained a five-layer FNN and trained for 500 epochs with the learning rate $10^{-3}$ on each meta-test graph. GCN is comprised of two graph convolutional layers and three linear output layers. It was trained for 500 epochs with the learning rate $10^{-3}$ on each meta-test graph. GAT is an extension approach of GCN, which implements the attention mechanism on message-passing operations. We trained GAT on each test graph for 500 epochs with the learning rate $10^{-3}$.

## 5 Results

### 5.1 Performance

Table 3 presents the results of the experiments, which includes the mean AUC score along with the standard deviation across 10 times of experiments. The proposed method, HGML, has superior performance compared to the other comparative methods. DSGCN does not surpass the proposed HGML because it cannot use edge information for encoding attributes. In contrast, HGML's attribute-wise message-passing network simultaneously considers node attributes and neighbor information. This allows it to learn enhanced attribute representations. MAMLGCN has the worst performance among all approaches because it cannot obtain node embeddings with attribute information from heterogeneous attribute graphs, and there is insufficient edge data to learn the edge patterns from various graphs. The three non-meta-learning models cannot reach better performance because they cannot learn from additional graphs. While NN performs better than GCN and GAT at the support rate $r = 0.3, 0.5$, GCN and GAT outperform NN when more support edges are available under the setting of $r = 0.7$. GAT did not outperform GCN due to little information in the training graphs to support its more complex model structure.

Table 4 presents the number of parameters, average training time for meta-learning, and inference time. The inference time refers to the time required for the model to process a query graph and generate predictions for the query edges, including the local adaptations of MAMLGCN and the re-training required for NN, GCN, and GAT. To evaluate inference time, we sample three different graph sizes: $N = 500, 5,000, 20,000$ from Reddit, ensuring that each sampled graph contains $N$ nodes and $N$ edges to simulate the few-shot scenario. HGML achieves better performance while maintaining similar model complexity and training time compared to DSGCN and MAMLGCN. MAMLGCN and the non-meta-learning approaches have higher inference time. In contrast, HGML can be applied to new graphs with a reasonable inference time.

Table 4: The average training time for meta-learning, inference time on graphs in size of $N$, and the number of parameters of HGML and the comparable methods.

| Model | # Parameters (K) | Training time (hours) | Inference times (ms) | | |
|-------|------------------|-----------------------|----------------------|--------|--------|
| | | | $N = 500$ | $N = 5,000$ | $N = 20,000$ |
| HGML | 23.2 | 11.1 | 24 | 25 | 104 |
| DSGCN | 29.2 | 13.8 | 3 | 33 | 22 |
| MAMLGCN | 20.9 | 10.4 | 493 | 579 | 3,215 |
| NN | 21.2 | - | 840 | 840 | 840 |
| GCN | 21.2 | - | 1562 | 1563 | 1575 |
| GAT | 21.6 | - | 4202 | 4204 | 4215 |

Table 5: Performance of HGML using different numbers of datasets for training.

| #Datasets for training | $r = 0.3$ | $r = 0.5$ | $r = 0.7$ |
|------------------------|-----------|-----------|-----------|
| 2 | 0.623±0.035 | 0.663±0.042 | 0.724±0.043 |
| 5 | 0.621±0.035 | 0.678±0.026 | **0.730±0.033** |
| 8 | **0.633±0.031** | **0.687±0.025** | 0.719±0.050 |

We evaluated HGML's ability to learn from diverse datasets by reducing the number of training datasets. The results are listed as Table 5. When we used all eight datasets for training, the model performed better under the support rate $r = 0.3, 0.5$. At the support rate $r = 0.7$, the results demonstrate that using 5 datasets for training may have better performance. This is likely because the model performs well with certain specific combinations of training datasets relative to the test datasets. However, using more datasets for training HGML generally results in better performance. Although we used data from various fields, they have some similar graph structures, and meta-learning can identify and learn useful common patterns shared among them. Unlike previous meta-learning methods, our model is not constrained by heterogeneous attribute spaces across graphs. In our experiments, we demonstrated that adding more graphs to the training set leads to better performance, even with a high proportion of SNS graphs in the data. We will test the model by adjusting the proportion of each category to better understand the impact of similarity between training and testing graphs.

Besides the neural network-based approaches described in this section, we also applied classic graph completion algorithms, such as Singular Value Decomposition (SVD), to our prediction tasks. However, the performance was so close to random guessing that we chose not to compare it with our approach.

## 5.2 Ablation study and hyper-parameters analysis

We perform ablation studies to verify the effectiveness of each element in HGML. The result is listed as Table 6. (a) we removed the message-passing from attribute-wised message-passing networks so that the attribute-specific node embeddings are obtained without using the information of neighboring nodes:

$$\mathbf{v}_{ni}^\ell = \begin{cases} f_{\mathrm{v}}^\ell(x_{ni}) & \text{for } \ell = 1 \\ f_{\mathrm{v}}^\ell([\mathbf{v}_{ni}^{\ell-1}, \mathbf{u}_n^{\ell-1}]) & \text{for } \ell = 2, \ldots, L. \end{cases} \tag{6}$$

The model failed to learn the pattern of the edges, and the prediction results are close to random guesses. (b) we trained HGML by minimizing the binary cross-entropy loss, which is commonly used in existing research for graph edge prediction, rather than maximizing the smoothed AUC as proposed in our method. The results demonstrate that using the proposed smoothed AUC as the loss can reach better performances under all three support rate setups.

In the hyper-parameters analysis, we compared the models with different encoder layers, hidden units, and the ratio of negative samples. First, we analyze the effect of the encoder layer $L$. The results are demonstrated in Figure 3(a). Under the support rate $r = 0.3$, performance did not significantly improve as we increased

Table 6: Ablation study.

|  | $r = 0.3$ | $r = 0.5$ | $r = 0.7$ |
|---|---|---|---|
| HGML | **0.633±0.031** | **0.687±0.025\*** | **0.719±0.050\*** |
| (a)Without message-passing operation | 0.518±0.014 | 0.517±0.017 | 0.521±0.012 |
| (b)Trained using binary cross-entropy loss | 0.616±0.018 | 0.622±0.039 | 0.656±0.025 |

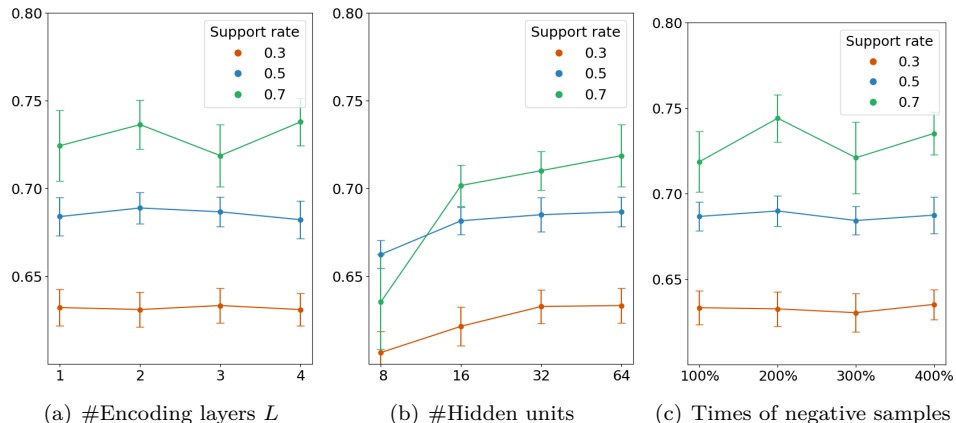

(a) #Encoding layers $L$      (b) #Hidden units      (c) Times of negative samples

Figure 3: AUC results for the experiments under different setups.

the number of encoding layers. However, when $r = 0.5, 0.7$, the model performed better using two and four encoding layers. This suggests that additional encoder layers might have greater potential to learn more complex patterns in specific situations. Next, we adjust the hidden units of neural network components $f_v^\ell$. The results are demonstrated in Figure 3(b). The model performed better when increasing the hidden units of the neural networks under all three support/query settings. Under the setting of support rate $r = 0.7$, the model required more hidden units to achieve better results due to the increased information in the input graphs. Finally, we verified HGML by adding more negative samples for training. In the original setup for training HGML, the number of negative samples matched the number of support edges for calculating the loss and gradients. We increased the negative samples to two to four times the support edges in this experiment. The results are demonstrated in Figure 3(c). Similar to the experiments of adjusting the encoding layers, the performances did not change significantly when $r = 0.3, 0.5$. However, under the support rate $r = 0.7$, the models performed better when we increased the amount of negative samples.

## 6 Conclusion

In this research, we proposed a novel approach, HGML, that addresses the few-shot edge prediction problem for graphs. The proposed attribute-wise message-passing networks can be shared and learn common patterns from graphs with heterogeneous node attributes. The proposed meta-learning algorithm allows the model to learn from various subgraphs and apply them to new test graphs with sparse edge data. In our experiments, HGML outperformed the existing approaches on 14 real-world graph datasets. HGML has the potential to be applied to many real-world applications. For instance, it can be used to build prediction models on newly launched SNS or when data collection is costly, such as discovering protein-protein interaction. HGML can be trained with publicly available data regardless of differences in data attributes.

HGML can be extended in several directions. First, HGML is developed based on basic graph convolutional networks (Kipf & Welling, 2017). Incorporating recent advanced approaches could further improve performance. Second, the current model can only handle graphs with a single node type. Extending the model to deal with graphs with multiple types of nodes could enhance its flexibility for real-world applications. Third, we aim to evaluate the proposed method using more graphs from various sources. Further testing

on more diverse datasets will be part of future work. Fourth, while the combination of multiple non-linear layers and average pooling enables the model to learn from diverse graphs, it also reduces interpretability. Finally, due to subgraph re-sampling for each epoch and graph optimization, training HGML is computationally expensive. Efficiently searching for training graphs may shorten training time and make HGML more robust.

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
