# OpenReview forum: "Meta-Learning for Graphs with Heterogeneous Node Attribute Spaces for Few-Shot Edge Predictions"
_TMLR — Accepted by TMLR_

### Review · Reviewer_CJpF · 2024-10-18

**Summary Of Contributions:**

This paper proposed a novel meta-learning method that aims to solve two problems in previous graph meta-learning methods: (i) training graphs and test graphs share different attribute space; and (i) no known information about the test graph's information. Extensive experiments show the effectiveness of the proposed method.

**Audience:**

Yes

**Claims And Evidence:**

Yes

**Requested Changes:**

I have some questions.
- Is MAMLGCN (model-agnostic meta-learning) defined in 4.3 Comparative methods a method or a series of methods? In my opinion, it is a series of methods but it is only one line in your table which looks like just a single method. Besides, no citations are provided in 4.3 so I am a little confused. I am not sure what method it is.
- What is the difference between graph domain adaptation methods on link prediction and the proposed method?

**Strengths And Weaknesses:**

**Strengths**
-  This paper is well-written and easy to understand
-  The method is easy to follow.

**Weakness**
- The running time is much longer than other baseline methods. The scalability is not that good, especially on large-scale graphs. The performance gains aren't significant compared to the extra time.
- The chosen baselines seem out-of-date. The results are not that convincing.

---

> ### Author Response · Authors · 2024-11-12
>
> Dear Reviewer,
>
> Thank you very much for your insightful comments and constructive feedback. We appreciate the time and effort you have taken to review our manuscript and provide suggestions for improvement.
>
> ### **Response to weakness 1:**
> The main objective of this paper is to explore the potential for improving performance in the few-shot link prediction problem using graphs with nodes in heterogeneous attribute spaces. We first demonstrated that our model achieves good performance, and reducing the long training time will be one of our future work directions.
>
> ### **Response to requested changes 1:**
> MAMLGCN is a single method that applies MAML to GCN. We used a standard GCN as the prediction model and optimized the meta-parameters through a few task-specific local adaptations, following the original MAML approach.
> Since MAML requires all input data to share a common attribute space, we trained MAMLGCN without using any attributes for our research problem. We have updated the following description of MAMLGCN in section 4.3 for better clarity.
>
> > MAMLGCN is a single method that applies model-agnostic meta-learning (MAML) to GCN. We used a standard GCN as the prediction model and optimized the meta-parameters through a few task-specific local adaptations, following the original MAML approach.
>
> ### **Response to requested changes 2:**
> Graph domain adaptation methods primarily address distribution shifts between different prediction tasks, especially when prediction labels differ. These methods require that training and test graphs share the same node attributes, and test graphs are partially used during training. In contrast, our proposed method aims to capture common knowledge from graphs with diverse node attributes, allowing test graphs to remain completely unseen during training.
>
> Thank you once again for your valuable feedback, which has helped us improve the clarity and completeness of our manuscript.
>
> Best regards

---

### Review · Reviewer_91Y2 · 2024-10-28

**Summary Of Contributions:**

1. The authors propose a graph neural network-based model for graphs with nodes in heterogeneous attribute spaces.
2. They propose a meta-learning framework to improve edge prediction performance with sparse edge information without bilevel gradient-based optimization.
3. They show that the proposed method outperforms existing meta-learning methods for edge prediction using 14 real-world graphs with nodes in heterogeneous spaces.

**Audience:**

Yes

**Broader Impact Concerns:**

I have no concerns regarding the broader impact.

**Claims And Evidence:**

Yes

**Requested Changes:**

Please revise the paper based on the weaknesses above. Minor comments are given below:
- I’m not sure if we need Algorithm 1, which is straightforward, in the paper.
- In Section 4, the citations for datasets are in a wrong format.

**Strengths And Weaknesses:**

Strengths
1. Designing graph neural networks for heterogenous node features seems like an important problem.
2. More than 10 graph datasets were used for the experiments, demonstrating the superiority of the proposed method.
3. The average performance, standard deviation, and training time are well reported in the experiments.

Weaknesses
1. Meta-learning is an essential keyword of this work, but the proposed method does not seem like meta-learning. From my understanding, the loss is computed for each sample and then just summed. Where are meta-parameters of the proposed method?
2. I’m not sure why we need to sample subgraphs and use them in the experiments instead of using the original graphs. How can we guarantee that the sampled graphs preserve the original topological properties? Is the proposed approach applicable only with sampling?
3. The concepts of support and query edges seem to be from previous work (Huang & Zitnik, 2020), but not sufficient description is given in the paper. Please describe their roles and why we need them in more detail.
4. The proposed method does not require any meta-data for the features, e.g., their distributions, min, max, etc. How can this work? Does the data preprocessing play an essential role in this perspective?
5. The authors use the objective function which approximates the AUC score in a differentiable manner. There are many recent variants of Eq. (4) particularly for contrastive learning. I hope the authors can strengthen the discussion on the choice of an objective function, in relation to recent studies.

---

> ### Author Response · Authors · 2024-11-12
>
> Dear Reviewer,
>
> Thank you very much for your insightful comments and constructive feedback.
>
> **Response to weakness 1:**
> Unlike MAML-like meta-learning approaches, which learn initial meta-parameters for local adaptation, another meta-learning approach—based on encoder-decoder architectures exists-such as neural processes (Garnelo et al., 2018) and previous work (Iwata & Kumagai, 2020)-. Our proposed model follows this encoder-decoder-based approach, learning to predict new test tasks by optimizing a single set of parameters across diverse input graphs. We have revised Chapter 2 of the manuscript to illustrate the difference between these two styles of meta-learning methods and the position of our proposed approach.
>
> > In contrast to MAML-like meta-learning approaches, which learn initial meta-parameters for local adaptation, another meta-learning approach—based on encoder-decoder architectures uses neural networks to encode data, enabling the prediction of new test tasks by meta-learning a single set of parameters across diverse input graphs without requiring bilevel optimization. This kind of meta-learning approach optimizes the neural network to approximate the effect of fine-tuning.
>
> **Response to weakness 2:**
> We sampled subgraphs to train the model on a diverse set of graphs with varying topological properties and combinations (or orders) of node attributes. While sampled graphs may not fully retain the original topological properties, the model's purpose is to predict new, unknown graphs across diverse attribute domains. The model can also be trained without sampling; however, in that case, training graphs would need to exhibit diverse topological properties and node attributes. We revised section 4.3 to add more explanation of the reason for sampling subgraphs as training graphs.
>
> **Response to weakness 3:**
> The roles of support and query edges align with previous work (Huang & Zitnik, 2020). In both their work and ours, support edges are used to obtain node embeddings, while query edges are used to calculate the model’s evaluation loss. However, unlike previous work, which updates meta-parameters to task-specific parameters over multiple iterations, we update the model parameters—shared across all input graphs—in a single step. We have strengthened the description of the roles of support and query edges in Section 3.3.
>
> **Response to weakness 4:**
> Although we did not impose any specific conditions on the graphs used, we assumed an underlying common feature distribution across graphs. By using message propagation and pooling operations, we capture this common pattern in the input graphs. While we preprocess node features to ensure they fall within the range of 0 to 1, this does not alter their original distribution. Through meta-learning, the neural network learns the useful distribution of node attributes. We have revised the section 3.3 to improve the clarity:
>
> > Although we did not impose specific conditions on the input graphs, we assumed an underlying common feature distribution across them. The proposed meta-learning algorithm in HGML enables the model to meta-learn these common patterns from the input graphs.
>
> **Response to weakness 5:**
> If I understand correctly, the contrastive learning approach you mentioned refers to objective functions that use both positive and negative edges in the graph, such as binary cross-entropy and other pairwise loss functions mentioned in (Zhu et al., 2022). While these objective functions could also work for our model, it has been shown that objective functions designed to minimize the misclassification error rate may not lead to the maximization of AUC (Cortes et al., 2003). Another recent research (Iwata and Yamanaka, 2019) proposed using the smoothed AUC for an imbalanced classification problem. Consequently, we propose using the smoothed AUC score as the objective function, as it directly relates to the AUC score and is more effective in improving model performance. We have enhanced the manuscript to discuss the choice of the objective function as follows:
>
> > Numerous objective functions, such as binary cross-entropy ... However, it has been shown that objective functions designed to minimize misclassification error may not necessarily maximize the AUC score ...
>
> **Response to requested changes:**
> We have revised the manuscript accordingly.
>
> **References:**
> 1. Garnelo, Marta, et al. "Neural processes." arXiv (2018).
> 2. Iwata, T., & Kumagai, A. Meta-learning from tasks with heterogeneous attribute spaces. NeurIPS (2020).
> 3. Zhu, Dixian, Xiaodong Wu, and Tianbao Yang. "Benchmarking deep AUROC optimization: Loss functions and algorithmic choices." arXiv (2022).
> 4. Cortes, Corinna, and Mehryar Mohri. "AUC optimization vs. error rate minimization." NeurIPS (2003).
> 5. Iwata, Tomoharu, and Yuki Yamanaka. "Supervised anomaly detection based on deep autoregressive density estimators." arXiv (2019).
>
> Thank you once again for your valuable feedback.
>
> Best regards

---

### Review · Reviewer_PM1y · 2024-10-29

**Summary Of Contributions:**

This paper proposed a meta-learning method for edge prediction that can learn from graphs with nodes in heterogeneous attribute spaces. The proposed model consists of attribute-wise message-passing networks that transform information between connected nodes for each attribute, resulting in attribute-specific node embeddings. The node embeddings are obtained by calculating the mean of the attribute specific node embeddings. The encoding operation can be repeated multiple times to capture complex patterns. The attribute-wise message-passing networks are shared across all graphs, allowing knowledge transfer between different graphs.

**Audience:**

Yes

**Claims And Evidence:**

Yes

**Requested Changes:**

see above

**Strengths And Weaknesses:**

Strengths:

1. The paper is written well and is easy to understand.

2. The studied problem is very important.

3. The results seem to outperform state-of-the-art.

Weaknesses:

1. could the authors try more ablations on the support edge rates, to test the robustness of the proposed algorithm?

2. Could the authors inlcude more recent meta-learning baselines for comparison? There are multuiple related papers working on this.

3. Could the authors try to compare with non-GNN related methods? There are some classific graph completion algorithms by low-rank approximation available in literature. It is better to observe the robustness of these staistical approaches in the few-shot setting.

4. Is it possible to estimate the edge probabilty by the cosine similarity?

---

> ### Author Response · Authors · 2024-11-12
>
> Dear reviewer
>
> Thank you very much for your insightful comments and constructive feedback. We appreciate the time and effort you have taken to review our manuscript and provide suggestions for improvement.
>
> **Response to Weakness 1:**
> We only conducted the experiments with support edge rates of 0.3, 0.5, and 0.7 because the results have demonstrated that the proposed method is more robust than other comparative approaches, especially when only a limited number of support edges are available.
>
> **Response to Weakness 2:**
> Although there are multiple related papers, different from ours approach, these approaches are not able to learn from graphs with heterogeneous node attributes. For example, (Hassani, 2022) proposed converting heterogeneous attributes to a common dimension using simple padding before inputting them into a shared neural network; however, this approach fails when the test graph has a different attribute dimension. Another study, (Lin, Mingkai, et al., 2023), transfers node attributes through task-specific neural networks and uses the barycenter as a common latent space. However, this approach is only feasible with limited training data, as the number of training parameters increases rapidly with more data, making it difficult to train effectively.
>
> **Response to Weakness 3:**
> We implemented the Singular Value Decomposition (SVD) approach for our tasks. However, the average AUC ROC scores for all tasks were close to 0.5, indicating that it was not effective. We have included the SVD results in Section 5.1 as follows:
>
> > Besides the neural network-based approaches described in this section, we also applied classic graph completion algorithms, such as Singular Value Decomposition (SVD), to our prediction tasks. However, the performance was so close to random guessing that we chose not to compare it with our approach.
>
> **Response to Weakness 4:**
> This is possible; however, when evaluating edge probabilities using cosine similarity, the binary cross-entropy loss function is more suitable. We experimented with both binary cross-entropy and inner product to estimate edge probabilities, but the proposed Euclidean distance consistently performed better across all settings.
>
> **References:**
> 1. Hassani, Kaveh. "Cross-domain few-shot graph classification." *Proceedings of the AAAI Conference on Artificial Intelligence.* Vol. 36. No. 6. 2022.
> 2. Lin, Mingkai, et al. "Multi-domain generalized graph meta learning." *Proceedings of the AAAI Conference on Artificial Intelligence.* Vol. 37. No. 4. 2023.
>
> Thank you once again for your valuable feedback, which has helped us improve the clarity and completeness of our manuscript.
>
> Best regards

---

### Review · Reviewer_qJhD · 2024-11-01

**Summary Of Contributions:**

The paper titled "Meta-Learning for Graphs with Heterogeneous Node Attribute Spaces for Few-Shot Edge Predictions" introduces a new meta-learning model aimed at improving edge prediction in graphs, specifically when available data is sparse and the nodes possess heterogeneous attributes. The authors propose the Heterogeneous Graph Meta-Learning (HGML) model, which features an attribute-wise message-passing network to transform information across nodes and generate embeddings tailored for diverse attribute spaces. This design enables HGML to share parameters across different graphs, promoting knowledge transfer and enhancing edge prediction accuracy, even when test graphs vary significantly from training ones.

1.) A graph neural network-based model that can share parameters among graphs with nodes in heterogeneous attribute spaces. This allows the model to learn from and make predictions on graphs with different types and numbers of node attributes.
2.) A meta-learning framework that improves edge prediction performance with sparse edge information by learning from various graphs without requiring bilevel gradient-based optimization. This enables the model to generalize to new unseen graphs with limited edge data.
3.) An attribute-wise message-passing network architecture combined with average pooling, which allows the model to process graphs with different numbers of attributes and learn relationships between attributes.
4.) The ability to make predictions on new input graphs without local adaptations, unlike some other meta-learning approaches that require fine-tuning.

**Audience:**

Yes

**Claims And Evidence:**

Yes

**Requested Changes:**

1.) Better Interpretability
2.) While HGML’s ability to predict without fine-tuning is positioned as a strength, this approach could be a double-edged sword. Without task-specific adaptation, the model may not fully optimize its performance on new graphs, especially those with highly distinct structures or attribute distributions from the training set. In classic meta-learning, some level of task-specific adaptation often enhances performance on new tasks, which HGML bypasses. Maybe I am understanding this wrong but I would like to understand the meta-learning aspect vis-a-vis task-specific fine-tuning.

**Strengths And Weaknesses:**

Strengths:
1.) Existing literature often limits meta-learning on graphs to homogeneous attribute spaces, restricting models to graphs where nodes share similar attributes. The HGML model in this paper, however, introduces attribute-wise message-passing networks that allow the model to process and learn from graphs with heterogeneous attributes.

2.) Prior models, including GCNs and GATs (Veličković et al., 2018), often depend on dense edge data for effective learning, making them less applicable to real-world scenarios where edge information may be sparse or costly to obtain. This paper’s focus on few-shot edge prediction addresses these constraints by enabling the model to generalize from sparse edge data. This few-shot learning approach relates to works like Huang & Zitnik (2020), which also aim to improve performance with limited data. However, HGML’s attribute-wise message-passing networks give it an edge in handling sparsity across varied attribute spaces.

3.) The paper supports its contributions with a robust experimental setup across 14 graph types, demonstrating HGML’s superior performance against both traditional GNNs and newer meta-learning models. This empirical validation not only supports the model’s effectiveness but also addresses gaps in existing literature where heterogeneous graph datasets have been underexplored.

Weaknesses:
1.) The model’s training time is notably longer than that of other comparative methods, especially due to repeated subgraph resampling and additional encoding layers. This may limit its practicality in real-time applications or settings with limited computational resources, as noted in the results where HGML took longer to train compared to simpler models like GCN and GAT

2.) Although the attribute-wise message-passing design could contribute to model interpretability, the paper does not explore or evaluate this aspect in depth. Without specific interpretability analyses, the practical advantage of interpretability remains speculative, missing an opportunity to provide insights into how attribute-specific embeddings contribute to edge predictions in heterogeneous graphs.

3.)Although the model performs well on a variety of mid-sized datasets, its suitability for much larger graphs (e.g., millions of nodes) is unclear. The computational demands of the attribute-wise message-passing and repeated encoding layers may become prohibitive as dataset size increases, potentially limiting the model’s scalability for very large graph applications like large social networks or web-scale knowledge graphs.

---

> ### Author Response · Authors · 2024-11-12
>
> Dear Reviewer,
>
> Thank you very much for your insightful comments and constructive feedback. We appreciate the time and effort you have taken to review our manuscript and provide suggestions for improvement.
>
> ### **Response to weakness 1:**
> The main objective of this paper is to explore the potential for improving performance in the few-shot link prediction problem using graphs with nodes in heterogeneous attribute spaces. We first demonstrated that our model achieves good performance, and reducing the long training time will be one of our future work directions.
>
> ### **Response to weakness 3:**
> We used standard Graph Convolutional Networks (GCN) as our attribute-wise message-passing network. However, GCNs are not suitable for extremely large graphs due to the computational expense of the message-passing operation. One of our future works will focus on applying other existing graph neural networks, such as GraphSAGE (Hamilton et al., 2017), to our model, which may allow it to handle larger graphs.
>
> ### **Response to requested changes 1:**
> To enhance link prediction performance for our research problem, we use multiple non-linear layers and average pooling, which reduces model interpretability. However, as shown in Table 5, our model effectively learns from diverse graphs. Since interpretability is essential for machine learning approaches, we plan to explore ways to improve it as part of our future work and have added this limitation to our manuscript in Section 6 as follows:
>
> > Fourth, while the combination of multiple non-linear layers and average pooling enables the model to learn from diverse graphs, it also reduces interpretability.
>
> ### **Response to requested changes 2:**
> The proposed approach is an encoder-decoder-style meta-learning method, similar to neural processes (Garnelo et al., 2018) and previous work (Iwata \& Kumagai, 2020). This style of meta-learning approximates the effect of fine-tuning by optimizing the neural network across various training tasks. To enhance clarity on this point, we have added the following description in Section 2:
>
> > In contrast to MAML-like meta-learning approaches, which learn initial meta-parameters for local adaptation, another meta-learning approach—based on encoder-decoder architectures uses neural networks to encode data, enabling the prediction of new test tasks by meta-learning a single set of parameters across diverse input graphs without requiring bilevel optimization. This kind of meta-learning approach optimizes the neural network to approximate the effect of fine-tuning.
>
> ### **References:**
> 1. Hamilton, Will, Zhitao Ying, and Jure Leskovec. "Inductive representation learning on large graphs." Advances in Neural Information Processing Systems 30 (2017).
> 2. Garnelo, Marta, et al. "Neural processes." arXiv preprint arXiv:1807.01622 (2018).
> 3. Iwata, T., \& Kumagai, A. (2020). Meta-learning from tasks with heterogeneous attribute spaces. Advances in Neural Information Processing Systems, 33, 6053-6063.
>
> Thank you once again for your valuable feedback, which has helped us improve the clarity and completeness of our manuscript.
>
> Best regards

---

### Review · Reviewer_Sn7r · 2024-11-25

**Summary Of Contributions:**

This research addresses improving the accuracy of link prediction for graphs with heterogeneous node attributes. The real-world graphs subject to edge prediction are very sparse and the number of training data is limited. To cope with the small amount of data, a meta-learning framework is introduced. Existing meta-learning frameworks target tasks where all node attributes are known during training and testing. Another existing approach is very time-consuming. The proposed method is capable of dealing with unknown node attributes. Through numerical experiments on 14 real-world datasets, the effectiveness and versatility of the proposed method is demonstrated.

**Audience:**

Yes

**Claims And Evidence:**

Yes

**Requested Changes:**

Please see the weaknesss above.

**Strengths And Weaknesses:**

## Strength

- The proposed method is a flexible framework that can be applied to graphs with heterogeneous node attributes. The proposed method can work when some node attributes are unknown during training and testing in reasonable runtime.
- Throughout the experimental results, the proposed method consistently shows higher AUCs than existing methods.
- Experiments have been conducted on real-world graphs. The experimental results suggest that the proposed method shows promising performance on real data, supporting its real-world applicability.

## Weakness

- The graphs used in the experiment covered SNS, Citation and EC, with a particularly high proportion of SNS graphs. As such graphs are known to be scale-free, the structure of the graphs used in the experiment is likely to be similar. Therefore, the effectiveness of proposed method for graphs with heterogeneous attributes and structures should be clarified in the future.
- When comparing the performance of the proposed and existing methods, do we need to consider differences in model size (number of parameters)? In a comparison of the runtime with the existing methods, the authors state that the proposed method requires a longer computation time than the existing methods due to the higher number of layers in the proposed method.

---

> ### Author Response · Authors · 2024-12-04
>
> Dear Reviewer,
>
> Thank you very much for your insightful comments and constructive feedback. We appreciate the time and effort you have taken to review our manuscript and provide suggestions for improvement.
>
> ### **Response to weakness 1:**
> Even when the types of graphs differ, meta-learning approaches remain effective as they can identify and learn useful common patterns shared across similar graph structures. However, previous meta-learning approaches (Huang et al., 2020) are limited by heterogeneous attribute spaces. Our experimental results in Table 5 demonstrate that models trained with more graphs, even those with a high proportion of SNS graphs, achieve better performance on test data comprising diverse graphs.
> We have revised our manuscript to enhance the description of learning from heterogeneous structures and the data proportions.
> >     Although we used data from various fields, they have some similar graph structures, and meta-learning can identify and learn useful common patterns shared among them. Unlike previous meta-learning methods, our model is not constrained by heterogeneous attribute spaces across graphs. In our experiments, we demonstrated that adding more graphs to the training set leads to better performance, even with a high proportion of SNS graphs in the data. We will test the model by adjusting the proportion of each category to better understand the impact of similarity between training and testing graphs.
>
>
> ### **Response to weakness 2:**
> As described in Section 4.2 of the manuscript, the number of parameters in our attribute-wise message-passing network is 32×3 (hidden units × layers), which is the same as our comparative method GCN. Since we repeatedly encode the embeddings three times, the total number of parameters increases threefold. The computation time for processing a graph in our model is approximately three times that of a standard GCN. To effectively learn common patterns from multiple graphs, our model requires a high number of training epochs. However, once trained, predicting a new graph using our model is fast, making it suitable for many real-world applications.
>
> ### **References:**
> 1. Huang, Kexin, and Marinka Zitnik. "Graph meta learning via local subgraphs." Advances in neural information processing systems 33 (2020): 5862-5874.
>
> Thank you once again for your valuable feedback, which has helped us improve the clarity and completeness of our manuscript.
>
> Best regards

---

### Decision · Action_Editor_ty7n · 2025-01-02

**Recommendation:** Accept with minor revision

**Comment:**

The authors did a good job with their explanations and they mostly addressed all the concerns that the reviewers had. Still, the recommendations are rather mixed, with one reviewer recommending a reject decision. I see the same issue the latter had: the fact that the proposed method requires significantly longer execution times and has significantly more complex architectures than other learning approaches that it was compared to. This leads to the question of fairness of comparisons, especially on larger graphs (such as knowledge graphs used in recommender systems and biomedical research, where heterogeneity may be an issue).

I am requesting that the reviewers compare methods in a way that the execution times, number of layers ( or model complexity) etc are all comparable. Otherwise, the utility of the method is rather mute. I also reserve the right to reverse my decision to reject if the comparisons are not adequately implemented. Out goal should always be to accurately investigate the pros and cons of all methods in an unbiased manner.

**Audience:**

This is definitely a topic of significant interest to the ML community and it is as timely as it is practically relevant.

**Claims And Evidence:**

The information presented in the manuscript, pertaining to the link prediction performance of various methods on 14 datasets, is accurate (as I can infer from the comments/questions of the reviewers). I am not as convinced about the utility of the newly proposed method due to the way comparisons were made.